# Autoimmune Hepatitis: A Diagnostic and Therapeutic Overview

**DOI:** 10.3390/diagnostics14040382

**Published:** 2024-02-09

**Authors:** Lydia A. Mercado, Fernando Gil-Lopez, Razvan M. Chirila, Denise M. Harnois

**Affiliations:** 1Department of Liver Transplant, Mayo Clinic Florida, Jacksonville, FL 32224, USA; 2Department of General Internal Medicine, Mayo Clinic Florida, Jacksonville, FL 32224, USA; chirila.razvan@mayo.edu; 3Department of Gastroenterology and Hepatology, Mayo Clinic Florida, Jacksonville, FL 32224, USA

**Keywords:** autoimmune hepatitis, immunosuppression, liver disease, autoantibodies, remission

## Abstract

Autoimmune hepatitis is an immune-mediated inflammatory condition of the liver of undetermined cause that affects both sexes, all ages, races, and ethnicities. Its clinical presentation can be very broad, from having an asymptomatic and silent course to presenting as acute hepatitis, cirrhosis, and acute liver failure potentially requiring liver transplantation. The diagnosis is based on histological abnormalities (interface hepatitis), characteristic clinical and laboratory findings (increased aspartate aminotransferase, alanine aminotransferase, and serum IgG concentration), and the presence of one or more characteristic autoantibodies. The large heterogeneity of these clinical, biochemical, and histological findings can sometimes make a timely and proper diagnosis a difficult task. Treatment seeks to achieve remission of the disease and prevent further progression of liver disease. First-line therapy includes high-dose corticosteroids, which are later tapered to decrease side effects, and azathioprine. In the presence of azathioprine intolerance or a poor response to the standard of care, second-line therapy needs to be considered, including mycophenolate mofetil. AIH remains a diagnostic and therapeutic challenge, and a further understanding of the pathophysiological pathways of the disease and the implementation of randomized controlled trials are needed.

## 1. Introduction

Autoimmune hepatitis (AIH) is a clinical syndrome produced by an immune-mediated inflammatory condition of the liver with an unknown etiology [1]. The AIH age spectrum is broad, from small infants to adults >60 years old. It has been described in both sexes (more common in women), all races and ethnicities [2]. AIH lacks pathognomonic features, requiring both the presence of characteristic histologic abnormalities, the biochemical elevation of aspartate aminotransferase (AST), alanine aminotransferase (ALT) and total IgG, and the presence of one or more circulating antibodies to establish diagnosis [3]. The clinical presentation of AIH can range from asymptomatic disease to acute liver failure, sometimes requiring liver transplantation. AIH can be associated with other liver conditions, particularly other auto-immune and inflammatory cholestatic liver diseases, such as primary biliary cholangitis (PBC) or primary sclerosing cholangitis (PSC). Furthermore, it may overlap with other entities, such as drug-induced liver injury (DILI), alcoholic or non-alcoholic steatohepatitis (NASH), viral hepatitis, or it may appear de novo in liver transplant (LT) recipients [1].

An adequate and timely diagnosis allows an early start of immunosuppression, thus preventing the progression of liver injury. Ideally, treatment aims at the normalization of aminotransferases and IgG (biochemical remission) within six months and the maintenance of remission thereafter [4]. Different therapeutic approaches have been established and are carefully chosen in an individualized manner. The first-line treatment has been established as steroids +/− azathioprine. However, when remission is not achieved or significant side effects are present, second-line treatments need to be considered [1,3]. LT is the last therapeutic resource in patients with liver failure. The diagnosis of AIH remains to this day a challenging task, considering the heterogenous clinical manifestation and broad age of presentation.

## 2. Epidemiology

AIH occurs within all ages and ethnic groups, and, interestingly, its clinical expression seems to vary depending on race and ethnicity. AIH is more predominant among adult females (71–95%) [5] and girls (60–76%) [6]. Age peaks have been typically described at 10–30 and 40–60 years [7]; nonetheless, in recent years, a later peak of onset has been reported almost universally, mainly considering Western Europe and North American populations, in >60 years old individuals. [8,9,10]. At disease onset, a high frequency of icteric disease seems to be present in Alaskan natives, and more severe disease is present in North American Aboriginal/First Nations [11]. In the Hispanic population with AIH, belligerent biochemical and histological features are often seen as well as a high prevalence of cholestatic disease [12]. Cirrhosis is commonly found among Hispanic [13] and African American populations, with accelerated progression of the disease. The latter is a group that has also shown a high rate of recurrence after LT, along with a higher mortality than white American patients [14]. The incidence of AIH varies worldwide and mainly depends on the region and age at onset. A marked increase in incidence of nearly 50% has been reported in recent decades in Denmark, Sweden, Spain, and the Netherlands [5,7,15]. Moreover, recent large epidemiological studies including populations from five continents (although including mostly western world countries) have reported changes in incidence and prevalence, with AIH affecting mainly countries with a high human development index, adults aged over 65 years old, high latitude (>45 degrees), and North American populations [9,10].

### Genetic Susceptibility

HLA A1-B8-DR3 and HLA DR4 haplotypes have long been strongly linked to AIH, with 84% of AIH patients from Western Europe and North America having these haplotypes. Moreover, patients from Japan (DRB1*0405), Mexico (DRB1*0404), Argentina (HLA-DRB1*0301, DRB1*0405) and Brazil (DRB1*13 and DRB1*03) have a particular HLA risk profile [16,17]. Also, the A1-B8-DR3 haplotype has been linked to more severe disease, the need for LT, a reduced response to steroids, and more relapse after transplant [18].

The differences seen across populations might reflect genetic predispositions, along with different immunologic and environmental influences [3]. However, it is highly likely that socioeconomic factors (delivery of healthcare, delayed diagnosis, competing risk factors) also play an important role in the pathogenesis.

## 3. Clinical Spectrum

### 3.1. Asymptomatic Disease

The heterogenic nature of AIH causes a broad spectrum of clinical manifestations. It has been reported that about 25–34% of patients with AIH are asymptomatic; however, histological findings may be similar to symptomatic patients [19]. A decreased 10-year survival has been reported in untreated asymptomatic patients compared to treated patients with more severe disease (67% versus 90%) [19]. The development of symptoms in this population can happen within 2–120 months (mean 32 months) [20], but the nonappearance of symptoms should not delay treatment. Final diagnosis is often established after an investigation of unexplained elevated serum aminotransferases during routine testing. At diagnosis, about one third of adults and half of children will have advanced liver disease, with the presence of cirrhosis [5,21].

### 3.2. Symptomatic Disease

In symptomatic patients, easy fatigability is the main complaint (85%), and jaundice can be present [3]. Chronic non-specific symptoms like malaise, arthralgias and amenorrhea often appear. As mentioned previously, in about one-third of patients, cirrhosis has developed by the time of diagnosis and physical signs of chronic liver disease may have appeared (splenomegaly, spider nevi, palmar erythema, caput medusa). In advanced stages, signs of portal hypertension such as ascites, esophageal varices, portal hypertensive gastropathy, cytopenias and hepatic encephalopathy can be seen [1]. Although less common than in other chronic liver diseases, hepatocellular carcinoma (HCC) can develop in the context of cirrhosis, and surveillance is recommended [1]. Symptoms of depression and anxiety are not uncommon and should be properly addressed to improve the health-related quality of life of these patients [22].

AIH presents with an acute onset in 25–40% of patients [23,24]. Further details of this entity will be discussed in the next section.

## 4. Specific Clinical Features and Presentations

In patients with AIH, clinical, biochemical, serological and/or histological features of PBC or PSC may present, or vice versa; in patients with PBC or PSC, AIH may occur. These are called “overlap syndromes”. Those entities as well as other AIH-specific scenarios are discussed in the following sections.

### 4.1. AIH and PBC

The prevalence of AIH–PBC is about 8–10% in adult patients with either PBC or AIH. The most validated criteria used to define this syndrome are the “Paris criteria”, with a sensitivity of 92% and a specificity of 97% [25]. For PBC, two of the following three criteria should be met: (1) serum alkaline phosphatase level ≥2-fold the upper limit of normal (ULN), (2) presence of antimitochondrial antibodies (AMA), and (3) florid bile duct lesion upon histologic examination. Criteria for AIH in PBC: In addition to interface hepatitis (mandatory), (1) serum ALT level ≥5-fold ULN and (2) serum IgG level >2-fold ULN or the presence of anti-smooth muscle antibodies (ASMAs) [26,27]. This variant is to be considered whenever a patient with PBC is non-responsive to treatment with ursodeoxycholic acid (UDCA), and the need for immunosuppression should be assessed [28].

### 4.2. AIH and PSC

Features of coexisting AIH and PSC have been described in both adults and children [29,30]. However, the diagnostic criteria for this condition are not well-defined. An estimated prevalence of 7–14% is presumed [27]. PSC diagnosis should be established based on typical cholangiographic findings (large duct PSC or small-duct PSC based on “onion skinning” periductal fibrosis upon histology) [31]. AIH-PSC diagnosis is made in the setting of overt cholangiographic or histological features of PSC, together with biochemical, histological, and serological features of AIH. Lesions of both AIH and sclerosing cholangitis have been described in children with AIH, and the term “autoimmune sclerosing cholangitis” (AISC) has been introduced [29]. The investigation of the biliary tree with magnetic resonance cholangiopancreatography (MRCP) in children with AIH is therefore suggested [29,32]. About 42% of adult patients with AIH and ulcerative colitis (UC) have PSC [33], as well as 45% of children with AISC [34]. Treatment with immunosuppression seems to be required [35].

### 4.3. DILI and AIH

DILI can mimic AIH, as both are mediated by specific immune reactions in hepatocytes, and clinical and histological overlap may happen. Hypersensitivity drug reactions have been reported in patients with classical features of AIH (2–17%) [36,37]. Nitrofurantoin [36,38], minocycline [36,39], and infliximab [40] are the most commonly associated drugs. The latency period from drug exposure to disease onset can range from 1–8 weeks to 3–12 months [41]; however, both nitrofurantoin and minocycline exceed 12 months [42]. The histological distinction between DILI and AIH remains a challenge, as DILI can present the characteristic histological features of AIH, except for fibrosis and cirrhosis [43,44,45]. The frequency of drug-induced AIH-like syndrome accounts for 9–12% of cases presenting with classic features of AIH [36,46]. History of exposure to a drug that may cause DILI is of major importance [36]. The primary treatment for DILI is drug withdrawal, but in some cases, DILI is also treated with steroids. The differentiation between DILI and AIH can only be achieved prospectively: in DILI, steroid discontinuation does not cause relapse, while in AIH, relapse will occur within a few months of steroid discontinuation. The close monitoring of steroid tapering and possible withdrawal is advised [1].

### 4.4. AIH and Pregnancy

In patients with known AIH, amenorrhea and reduced fertility occur when the disease is poorly controlled [3], whereas effective immunosuppression has enabled the occurrence of pregnancy in women of childbearing age [1]. AIH is uncommonly diagnosed during pregnancy but may present in the post-partum period (three times more common) [47]. In pregnant women with AIH, the live birth rate is 73%, and the rate of fetal loss and stillbirth is 27% (7–15% in general population) [48]. Premature births occur in approximately 20% of pregnancies. The overall rate of maternal complication is 38% during pregnancy or within 12 months of delivery [49]. Patients with established cirrhosis are at the greatest risk of adverse outcomes during pregnancy and in the first-year post-partum [48].

### 4.5. Viral Hepatitis and AIH

Even though viral hepatitis should be excluded prior to establishing a diagnosis of AIH, AIH may occasionally develop in patients with hepatitis B virus (HBV) or hepatitis C virus (HCV) infection, and, of course, patients with AIH can contract viral hepatitis, infections to which these patients are more susceptible [50]. Studies performed in patients with HCV have reported the development of AIH following treatment with interferon-alpha [51,52]. The immunostimulatory side effects of interferon-alpha have made the differentiation between AIH and HCV a challenge, but owing to the advent of interferon-free drug regimens, this no longer represents an issue [1]. In cases of diagnostic uncertainty, HCV can be treated first; if inflammation persists, the diagnosis of AIH should be considered and treated with adequate immunosuppression. Baseline HBV serologies are recommended for patients receiving immunosuppression, as the reactivation of the virus has been reported during the treatment of AIH [1]. Vaccination against Hepatitis A virus (HAV) and HBV is encouraged in patients with AIH, ideally before starting immunosuppression [1].

### 4.6. De Novo AIH after LT

De novo AIH refers to the development of AIH in patients who have received a transplant for liver diseases other than AIH [53]. It is important to differentiate de novo AIH from plasma cell hepatitis/rejection. The features of de novo AIH are similar to those required for diagnosis of AIH and recurrent AIH [54]. In pediatric British children who underwent LT for extrahepatic biliary atresia, Alagille syndrome, drug-induced acute liver failure and alpha 1 antitrypsin deficiency, AIH was reported in 4% of patients 6–45 months after transplant [55]. The incidence of AIH in transplanted adults is 1–3%, with an overall incidence of 4 cases per 1000 patients; this has been described particularly in patients transplanted for PBC [56] or HCV [57]. A timely diagnosis and recognition can prevent graft rejection and improve survival [1].

### 4.7. Acute Disease, Acute Liver Failure and Chronic Complications in AIH

AIH is one of the most common causes of acute liver failure (ALF) in the USA. According to the AASLD, acute severe AIH is defined as an acute liver injury (ALI) with jaundice plus INR > 1.5 and <2.0 [3]. As mentioned earlier, acute AIH is a frequent form of presentation [24]. Around 6% of patients develop ALF according to data from European and North American patients [58]. Recent data from a US cohort reported that 16% of patients with AIH-associated ALI subsequently developed ALF. At presentation, the concentration of ALT was 449 (median, 227–805) and that of bilirubin was 22.8 (median, 17.9–28.0). Only 70% had antinuclear antibodies (ANAs) ≥ 1:40 and 43% had anti-smooth muscle antibodies (ASMA) ≥ 1:40. At 21 days, transplant-free survival was 15% and 24% died without liver transplant.

In the acute AIH group, approximately 11% of patients were reported as negative for ANA and ASMA, and immunoglobulin levels were also found normal in 15–39% [59,60]. Moreover, the simplified AIH diagnostic score has been shown to be less sensitive than the original revised version in this group [61].

Liver biopsy should be performed if possible. The histopathologic findings in this context might include lobular hepatitis, lymphoplasmacytic infiltrate, and interface hepatitis in support of acute AIH. The specific features of ALF encompass central perivenulitis, lymphoplasmacytic infiltrate, lymphoid follicles, and massive hepatic necrosis [58]. Heterogeneous hypoattenuated regions within the liver can be seen in unenhanced computer tomography in 65% of patients with acute severe AIH [62]. Thus, this population remains a diagnostic challenge due to discrepancies in the laboratory and histological phenotypes compared to classical AIH.

Chronic complications of AIH, as in any other chronic liver disease, include cirrhosis with the risk of developing HCC (although less frequent than other causes of cirrhosis) [5,63]. It has been shown in several studies that male gender and the presence of cirrhosis contribute greatly to the development of HCC, at a rate of 1–2% per year [56]. Liver ultrasonography every six months for the surveillance of HCC in AIH appears reasonable in patients with cirrhosis. Complications associated with long-term immunosuppression need to be contemplated. Patients with AIH seem to be at an increased risk of higher extra-hepatic malignancies according to a population study, occurring in 5%; non-melanoma skin cancers are the most common [64,65].

## 5. Diagnostic Workup

The diagnosis of AIH can be established when characteristic histologic abnormalities (interface hepatitis), clinical and laboratory findings (elevated AST, ALT, and serum IgG) are present, along with one or more circulating antibodies. Unfortunately, there is no pathognomonic marker of AIH, and its diagnosis requires the exclusion of other liver diseases that may resemble it. Figure A2 depicts a proposed initial approach.

### 5.1. Biochemical Findings

The typical biochemical profile in AIH is predominantly a hepatocellular pattern, with bilirubin and aminotransferases concentrations ranging from just above ULN to >50 times these levels. Cholestatic enzymes are usually normal or slightly increased [66,67]. Increased levels of gamma-glutamyl transferase (GGT) can also be seen in AIH, but not alkaline phosphatase (ALP) typically [68,69]. It is relevant to mention that the degree of ALT elevation does not reliably correlate with the severity of AIH at a histological level. In some cases, aminotransferases and GGT may normalize despite persistent inflammatory activity at a histological level. Spontaneous apparent biochemical remissions can, in many cases, delay and/or underestimate the diagnosis, which may explain the presence of cirrhosis in about one third of patients at the time of AIH diagnosis [1]. Increased GGT and IgG are found in 85% of patients with AIH [68], and in 25–39% at acute onset of the disease. High IgG levels are very characteristic of AIH, while increased IgA and IgM levels suggest a different diagnosis [70]. Full biochemical remission is the normalization of transaminase levels and IgG levels [71].

### 5.2. Autoantibodies

Autoantibodies are the hallmark of AIH and play a major role in diagnosis. There are essentially two types of AIH, based on the antibodies present. Type 1 AIH is characterized by the presence of ANA and/or ASMA/anti-actin antibodies (a subset of ASMAs). Type 2 is characterized by antibodies of liver kidney microsome type 1 (anti-LKM1), normally in the absence of ANAs and ASMAs [26]. Seronegative AIH is suspected in cases that are negative for ANAs, ASMAs and LKM1 antibodies, accounting for 20% of the cases [3]. ANAs are detected in 80% of adults in North America, ASMAs in 63%, and anti-LKM1 antibodies in 3%. About 49% of patients have either ANAs, ASMAs, or anti-LKM1 antibodies present at the time of AIH diagnosis, and the remaining 51% have multiple antibodies present [72]. None of the autoantibodies are disease specific for AIH and may be present in other liver diseases, which is why a careful assessment is required. When seronegative AIH is suspected, other antibodies should be sought. The antibodies of soluble liver antigen (anti-SLAs) are present in 7–22% of patients with type 1 AIH, having a 99% specificity for the diagnosis [73,74,75]. Anti-SLAs are the sole markers of AIH in 14–20% of patients and are associated with both severe disease and disease relapse after immunosuppression withdrawal. Atypical perinuclear antineutrophil cytoplasmic antibodies (pANCAs) can also be identified in type 1 AIH in 50–92% of patients [76,77]. Antibodies against filamentous (F) actin (antiactin) are a subset of ASMAs and are present in 86–100% of patients with AIH and ASMAs [78,79]. Interestingly, it has been shown that the detection of anti-F-actin antibodies is more consistently achieved through indirect immunofluorescence (observer’s experience-dependent) compared to the ELISA assay [80].

The antibody of anti-actinin (anti-α-actinin) can be found in 66% of patients with anti-actin; this dual reactivity is associated with a worse treatment response [81]. Anti-LKM3 antibodies are present in 17% of patients with type 2 AIH [82]. The antibodies of liver cytosol type 1 (anti-LC1) can be found in 32% of patients with anti-LKM1 and are associated with children and/or severe liver disease [83].

The main technique used for antibody testing is indirect immunofluorescence (IFL) [84]. It is performed on a freshly frozen rodent substrate that includes the kidney, liver and stomach and allows the detection of ANAs, ASMAs and anti-LKM1. Uncommon antibodies such as anti-LC1 and anti-LKM3 can also be identified in the absence of anti-LKM1 [84]. IFL does not detect anti-SLA/LP antibodies [84,85]. ELISA and immunoblotting are the only diagnostic tests used for anti-SLA/LP, but they may also identify anti-LKM1, anti-LKM3, and anti-LC1 [85]. The detection of autoantibodies is of crucial importance to the diagnosis, but unfortunately, the complete work-up for autoimmune serology is often not available in all laboratories. An increase in training and expertise is essential for laboratory personnel and clinicians to perform a correct interpretation of the results.

### 5.3. Histology

A liver biopsy with a compatible histologic finding of AIH is required to establish diagnosis and should be performed before treatment is started (unless contraindicated) [67,70]. In patients with an acute/fulminant onset of AIH, coagulopathy may be present; approaches such as transjugular biopsy or biopsy under control by mini laparoscopy have been proven safe. The typical hallmarks of AIH include interface hepatitis (at the portal parenchyma), along with plasma cell infiltration in 66% and lobular hepatitis in 47% [86]. Centrilobular necrosis can be present in 29% [87] of patients, despite the presence or absence of cirrhosis [88]; emperipolesis (penetration of one intact cell into another intact cell) is seen in 65% of patients [89], and hepatocyte rosettes appear in 33%, along with hepatocyte swelling and/or pycnotic necrosis [70]. Cirrhosis can be found in 33% of adult patients [90] and 38% of children [6]. Multilobular necrosis or bridging necrosis is seen in 48% of adults with cirrhosis. Unfortunately, none of these findings are unique for AIH, and other liver diseases are to be excluded; otherwise, concurrent diagnosis needs to be established. IgG4-positive plasma cells can sometimes be present, but the clinical impact remains unclear [91]. Findings consistent with nonalcoholic steatohepatitis can be present in up to 30% of patients with AIH [92]. The histological features that define AIH with ALF consist of central perivenulitis in 65%, plasma cell-enriched inflammatory infiltrate in 63%, massive necrosis in 42%, and lymphoid follicles in 32% [58]. The modified Histological Activity Index (mHAI) score helps the pathologist provide a quantitative evaluation of the inflammatory process [93]. Liver biopsy not only helps establish diagnosis but also can provide information on prognosis, management, and screening.

### 5.4. Diagnostic Scoring Systems

The International Autoimmune Hepatitis Group (IAIHG) created a diagnostic scoring system [94] that was later revised to aid in the diagnosis of AIH [67], and a simplified version was subsequently created [95]. The revised scoring system exhibits a greater sensitivity for AIH when compared to the simplified scoring system (100% versus 95%), while the simplified version has greater specificity (90% versus 73%) and accuracy (92% versus 82%) [96]. These score’s diagnostic performance has been tested in different geographic settings. Due to its clinical applicability, the simplified system was tested in various cohorts, including populations from Germany, Japan, Spain, Greece, Norway, Brazil, Austria, United Kingdom, and the USA, and validated in these populations by including viral, metabolic, toxic, cholestatic and genetic hepatic syndromes, with a sensitivity of 99% and specificity of 97% [70,95]. A validation study from a Chinese cohort reported that, for the simplified score, the sensitivity and specificity were 90% and 95%, respectively [97]. Likewise, an Italian study reported an overall sensitivity and specificity of 91% and 94%, respectively [98].

The revised criteria scoring system has been used in clinical practice to assess patients with few or atypical features of AIH [99]. However, although it is very extensive and complex, it fails to distinguish AIH from cholestatic syndromes.

Conversely, the simplified scoring system (Table 1) is based on four parameters: presence and titer of autoantibodies detected by IFL and ELISA (for anti-SLA/LP), concentration of serum IgG, presence of typical or compatible histology, and absence of viral hepatitis markers [70]. Although the simplified criteria work for AIH-PBC and are useful in excluding patients with AIH from patients with other conditions and concurrent immune features, they are more likely to exclude atypical cases of AIH [95] and fail to grade responses to corticosteroid treatment [100].

Given the aforementioned considerations, clinical judgment is crucial for the use of these scoring systems.

### 5.5. Noninvasive Assessment of Hepatic Fibrosis

#### 5.5.1. Biomarker Panels

A number of serum-based biomarker panels have been developed for the assessment of hepatic fibrosis. The most commonly used in the context of AIH are Fibrosis-4 index (FIB-4) [103], FibroTest [104,105], the enhanced liver fibrosis test, and the serum AST/platelet ratio index (APRI) [106]. Unfortunately, they are limited in terms of assessing the progression of hepatic fibrosis, reversal, prognosis, the risk of developing HCC, and treatment outcomes [3].

#### 5.5.2. Vibration-Controlled Transient Elastography (VCTE)

VCTE strongly correlates with the histological stage of fibrosis in patients with AIH. The best time to assess liver stiffness using VCTE is after at least 6 months of successful immunosuppressive treatment. The measurement is compiled from the presence of inflammation; thus, it should not be performed within the first few months of treatment being initiated [107]. VCTE can precisely diagnose cirrhosis and distinguish the different stages of cirrhosis (F0–F4) [107]. The cutoff values for the stages of cirrhosis are as follows: 5.8 kPa for F ≥ 2, 10.5 kPa for F ≥ 3, and 16 kPa for F ≥ 4 [107]. When assessed at least 6 months after treatment, a reduction in liver stiffness correlates with biochemical remission, the regression of fibrosis, and a better prognosis [3].

#### 5.5.3. Magnetic Resonance Elastography (MRE)

Like VCTE, the findings obtained with MRE correlate with the stage of fibrosis in the patient. Furthermore, the MRE evaluation of splenic stiffness may have a prognostic value for the prediction of portal hypertension and esophageal varices [108]. In patients with AIH, an accuracy of 97%, sensitivity of 90%, specificity of 100%, positive predictive value of 90%, and negative predictive value of 90% have been demonstrated in advanced hepatic fibrosis [109]. MRE has shown to be superior to conventional magnetic resonance, fibrosis scoring systems, and conventional laboratory tests for the diagnosis of cirrhosis in AIH [109].

#### 5.5.4. Acoustic Radiation Force Impulse Imaging (ARFI)

ARFI performs an assessment of liver stiffness by measuring the changes in the wave propagation speed; the displacements of short-duration bursts of radiated sound waves are interpreted as changes in liver stiffness [110]. ARFI has a proven accuracy of over 90%, with a sensitivity of 93% and a specificity of 85% [111]. Splenic stiffness measured with ARFI has also been correlated with the grade of esophageal varices [110]. The downside of ARFI is that it can overestimate hepatic fibrosis when features such as massive hepatic necrosis, cholestasis, severe inflammation, and hepatic congestion are present [112].

## 6. Therapy in AIH

In AIH, the goal of treatment is to achieve complete remission for the prevention of the further progression of liver disease, while limiting treatment-related complications. To accomplish this, permanent therapy is usually required, as only a minority of patients achieve sustained remission after treatment withdrawal. Czaja showed that the cumulative 10-year survival of untreated patients with AIH was less than that of treated patients (98% vs. 67%, *p* = 0.001) [20]. The decision to not treat a patient with AIH should be highly justified, especially in the presence of relative contraindications regarding the use of steroids. However, untreated AIH may have a shifting and unpredictable behavior, as asymptomatic patients may later become symptomatic, or the progression to cirrhosis with the potential development of HCC may occur [20]. Untreated patients should nonetheless be carefully monitored every 3–6 months, possibly with liver biopsy if their aminotransferases and/or IgG levels increase or fluctuate [1]. It is recommended that all patients with active disease AIH should receive treatment, and the dosage should be adapted to the activity of the disease [1].

### 6.1. Pre-Treatment Considerations

#### 6.1.1. Thiopurine Methyltransferase (TPMT)

TPMT is an enzyme involved in azathioprine metabolism. Patients with TPMT deficiency are at risk of azathioprine or mercaptopurine (6-MP)-associated toxicity and myelosuppression [113]. The 6-thioguanine nucleotides (6-TGN) are responsible for the immunosuppressive and anti-inflammatory properties of azathioprine, and unfortunately, they can also cause myelotoxicity. Undetectable TGN levels may reflect an altered metabolism or non-adherence to treatment, whereas high levels may suggest toxicity [3]. The frequency of intermediate TPMT deficiency in the general population ranges from 6–16%, and the occurrence of low TPMT is roughly 0.3–0.5% [114]. However, testing for TPMT activity does not reduce the appearance of other azathioprine or 6-MP symptoms like nausea, rash, arthralgias [115]. Patients with normal TPMT activity can still develop dose-dependent toxicities such as cytopenia, which are more commonly present in patients with cirrhosis [114]. Considering the potentially serious outcomes of azathioprine treatment in patients with TPMT deficiency, its use without a cost-effectiveness analysis is unwarranted. The close monitoring of patients with AIH receiving azathioprine is mandatory [1]. In patients with AIH and TPMT deficiency, monotherapy with prednisone, or a combination of lower-dose prednisone and mycophenolate mofetil (MMF), may be used [1].

#### 6.1.2. Vaccination

In patients with AIH, their vaccination status should be assessed prior to the start of immunosuppressive therapy [50]. Recombinant and inactivated vaccines are considered safe in patients on high doses of immunosuppression, and although response rates are lower, their use is still advised. Vulnerability to HAV and HBV has been reported in 51% and 86% of patients with autoimmune liver diseases, respectively [50]. Therefore, patients unprotected against HAV and HBV should be vaccinated prior to the start of AIH treatment [3].

#### 6.1.3. HBV Reactivation Detection and Prevention

Periodic serological testing, which includes HBsAg and HBV DNA, is suggested for patients with AIH who are HBsAg-negative/anti-HBc-positive while undergoing therapy with prednisone or prednisolone and azathioprine [3,116]. For patients with a high risk of reactivation due to their serological profile, type, dose, and/or duration of immunotherapy [116], prophylactic antiviral therapy with entecavir or tenofovir is recommended during immunosuppressive treatment and for at least six months after treatment withdrawal [3,116].

#### 6.1.4. Osteoporosis

The most common risk factors for osteoporosis are the prolonged use of glucocorticoids, postmenopausal status, age > 65 years for women and > 70 years for men, and a history of low trauma fracture [117]. Bone mineral densitometry with the dual-energy X-ray absorptiometry of the lumbar vertebrae and hips should be performed at baseline and every 2–3 years if ongoing risk factors are present [3,118]. For patients undergoing glucocorticoid treatment, elemental calcium (1000–1200 mg daily) and vitamin D (400–800 IU daily) are recommended [3], as well as bisphosphonates and weight-bearing exercise when osteoporosis is present [1].

#### 6.1.5. Metabolic Syndrome

Metabolic syndrome is a bundle of cardiovascular risk factors [119] diagnosed when at least three of the following are present: hypertension, hypertriglyceridemia, low high-density lipoprotein cholesterol level, fating hyperglyceridemia, and central obesity (waist/hip ratio or body mass index > 30 kg/m^2^) [120]. Prior to and during therapy in AIH, an assessment of all the features of metabolic syndrome should be performed. If diagnosed, individualized treatment, a potential reduction in the glucocorticoid dosage, and lifestyle modifications are advised [3,121].

### 6.2. First-Line Treatment

First-line treatment in AIH aims to improve patient symptoms, manage hepatic inflammation, accomplish biochemical remission (normalization of AST, ALT, and IgG) [122], prevent the progression of disease, and promote the regression of scarring in fibrosis and cirrhosis [1], while minimizing the risk of drug-induced complications. Therapy should be considered for all patients with active disease, as determined by clinical, biochemical, and histological assessment [3].

#### 6.2.1. AIH Induction Phase

During an induction phase in adults, prednisone alone (40–60 mg daily), or prednisone (20–40 mg daily) in combination with azathioprine (50–150 mg daily or 1–2 mg/kg daily in Europe), is administered along with an antiacid [1,3]. In the case that steroid monotherapy was initially preferred, azathioprine is administered 2 weeks after, once TPMT evaluation has been performed. For pediatric patients, prednisone alone (1–2 mg/kg with a maximum dose of 40–60 mg daily), or budesonide (9 mg daily) with azathioprine (1–2 mg/kg daily), is administered with an antiacid [3]. Prednisolone (1 mg/kg daily) is usually preferred over prednisone in Europe and is administered with weight-based doses of azathioprine (1–2 mg/kg daily). Laboratory testing is advised every 1–2 weeks. Once biochemical remission is achieved, the dose of prednisone or prednisolone is gradually reduced to 20 mg daily, or a sufficient dose to maintain biochemical remission while monitoring biochemical status every 2 weeks [3]. Gradual tapering is then recommended, with 2.5–5 mg every 2–4 weeks to achieve a dose of 5–10 mg dailyto maintain biochemical remission [3]. Then, prednisone (or prednisolone) may be fully discontinued, leaving the patient on azathioprine or alternative glucocorticoid-sparring drugs (especially in children) [3]. If a negative biochemical response is observed, diagnosis re-assessment should be performed, or second-line treatment should be considered. Prednisone monotherapy is an option for patients with AIH when treatment is expected to last for less than 6 months (suspected drug-induced AIH-like injury), or when azathioprine is contraindicated due to intolerance or complete TPMT deficiency [3]. In that case, mycophenolate mofetil (MMF) can be used as an alternative to maintain remission [3]. MMF has been used in lieu of azathioprine as a front-line therapy for AIH [69]. MMF in combination with prednisone in a single-center study reported a remission rate of around 75% after 24 months of treatment [123], but further data are needed to consider its implementation as a first-line treatment.

#### 6.2.2. AIH and Cirrhosis

In patients with AIH and cirrhosis, induction treatment with prednisone at 20–40 mg daily is advised for adults and 1–2 mg/kg daily is advised for children. In patients with decompensated cirrhosis, azathioprine should not be used. In the context of compensated cirrhosis, TPMT activity should be assessed and after 2 weeks; azathioprine can be initiated at 50–150 mg daily. During induction, laboratory testing should be performed every 1–2 weeks. The treatment response should be assessed at 4–8 weeks, and if biochemical improvements are seen, prednisone can be tapered to 5–10 mg daily over the following 6 months. If azathioprine was started, it should be maintained. If no improvements are seen in the patient’s biochemical profile, the diagnosis should be re-evaluated, and second-line drugs should be considered. Budesonide use is not advised in patients with AIH and cirrhosis. Portosystemic shunting in patients with cirrhosis can reduce drug efficacy and promote steroid-specific side effects by allowing budesonide to bypass the liver [124]. Portal vein thrombosis has also been reported in patients with cirrhosis on therapy with budesonide [125].

#### 6.2.3. Maintenance

Once biochemical remission is achieved, patients with AIH and patients with AIH-associated cirrhosis should undergo laboratory testing every 3–4 months [3]. Steroid withdrawal can be attempted while continuing azathioprine [3]. When prolonged biochemical remission is reached at 24 months, laboratory testing can be moved to every 4–6 months and immunosuppression withdrawal can be considered with or without the need for biopsy [3].

#### 6.2.4. Acute Severe AIH

In patients with acute severe AIH, prednisone at 60 mg daily should be started for adults and administered at 2 mg/kg daily, or IV steroids, for children. Budesonide and azathioprine should not be used in this population. Laboratory testing should be performed every 12–24 h [3]. The response to treatment needs to be assessed at 7–14 days, and if a biochemical response is seen, prednisone should be cautiously reduced. Azathioprine can be considered once cholestasis has resolved, but TPMT activity should be assessed first. If no biochemical response to treatment is seen, the diagnosis should be re-evaluated, and transplant evaluation initiated. If hepatic encephalopathy develops, urgent transplant evaluation is required. For maintenance therapy once biochemical remission is achieved, laboratory testing should be performed every 3–4 months [3]. The lowest dose of immunosuppression required to maintain remission should be indicated and should not be withdrawn.

Prolonged prednisone monotherapy at doses > 10 mg daily should be avoided, as they are associated with well-known cosmetic side effects (hirsutism, striae, facial rounding, dorsal hump), systemic impacts (glucose intolerance/diabetes, osteoporosis, fatty liver, hypertension), and a poor quality of life (psychosis, depression, anxiety) [126]. In patients receiving azathioprine, the presence of leukopenia or thrombocytopenia warrants dose reduction. If cytopenia, which is mostly found in patients with cirrhosis [127], does not improve within 1–2 weeks, the withdrawal of azathioprine is advised [3]. An azathioprine dose reduction or adjustments can be made to avoid toxicity, which can be monitored through thiopurine metabolite levels, while maintaining a therapeutic range. A summary of first-line treatment is depicted in Figure A1.

#### 6.2.5. Relapse in AIH

Relapse is defined by the IAIHG criteria as the reappearance of ALT elevation more than three times the ULN. It may also present with milder ALT elevations and/or an increase in IgG levels. Relapse after treatment withdrawal has been reported to occur in 50–87% of adults and 60–80% of children [122]. Therapy adherence should be assessed in non-responders to induction therapy or in those who relapse. Initial treatment with prednisone or prednisolone and azathioprine is recommended for relapse [1]. Permanent maintenance therapy is advised in patients with AIH after a relapse, although complete drug withdrawal has been possible in 12% of patients and it has been proposed that it can be attempted once the disease has been inactive for at least 24 months [128].

### 6.3. Second-Line Treatment

When the desired therapeutic response is not achieved after first-line therapy, either due to an incomplete response or drug intolerance (inability to continue treatment due to side effects), second-line treatment is to be considered and implemented in AIH [129]. Treatment failure has been reported in 7–9% of adults with AIH and is associated with an increased risk of the progression of liver disease to cirrhosis or liver failure [129]. Second-line therapies for AIH include MMF, calcineurin inhibitors (cyclosporin, tacrolimus), mercaptopurine (MP), and biologics (rituximab, infliximab) [3]. The partial normalization of aminotransferases and IgG is defined as an incomplete response and occurs in around 15% of patients with AIH, regardless of age [1]. A study performed in patients with type 1 AIH showed patients that responded to therapy within 24 months of treatment initiation had a lower frequency of progression to cirrhosis, unlike patients with a delayed response > 36 months [130]. Second-line therapy for incomplete response includes MMF and calcineurin inhibitors.

#### 6.3.1. MMF

In patients with AIH intolerant to azathioprine, or in the presence of an incomplete response or treatment failure (glucocorticoid/azathioprine), MMF can be used as a second-line option. In a metaanalysis, MMF showed an overall response rate of 58% [131]; the therapy was overall well tolerated, with a side effects rate of 14%. MMF discontinuation was possible in 8% of patients [132]. A study performed in patients treated with MMF due to failed standard therapy demonstrated remission in 60% of cases. The factors independently associated with a poor treatment response were a younger age, higher pre-treatment IgG levels, and higher INR. MMF proved to be more effective in patients with therapy intolerance (62%) than in patients with treatment failure (38%) [133].

#### 6.3.2. Calcineurin Inhibitors

In the setting of azathioprine intolerance, treatment failure, or incomplete response, tacrolimus has been used, with multiple studies confirming its moderate to high efficacy. Tacrolimus has been used in combination with prednisone, budesonide, azathioprine, or MMF, and with serum levels ranging from 1–10 ng/mL. Different studies have shown similar response rates, with the improvement and normalization of aminotransferases in 73% to 94% of patients with AIH receiving tacrolimus as second-line therapy [134]. The combination of tacrolimus with prednisone was the most effective for the normalization of aminotransferases [132].

#### 6.3.3. Liver Transplantation in AIH

The need for liver transplantation may emerge in the setting of acute onset AIH evolving into liver failure or end-stage liver disease. AIH is the indication for LT in 2% of patients in Europe [135] and up to 5% in the United States [136]. Acute rejection (50% versus 85.5%; *p* = 0.02) and steroid-resistant rejection (81% versus 46.8%; *p* =< 0.001) after LT are more common among patients transplanted for AIH when compared with patients transplanted for alcohol-related cirrhosis [137]. Patients transplanted for AIH (15.6%) also show a higher incidence of chronic rejection when compared to patients transplanted for PBC (8.2%; *p* = 0.002), PSC (5.2%; *p* < 0.05), or alcohol-related cirrhosis (2%; *p* < 0.001) [138]. Considering the higher frequency of rejection in patients transplanted for AIH, the continuation of glucocorticoid therapy after LT has been suggested in an attempt to protect against the recurrence of AIH and rejection. Although some patients may be safely weaned off corticosteroids, this remains controversial and further studies are required to evaluate safety.

### 6.4. Future of Biologic Therapy in AIH

AIH is considered refractory to the standard of care (steroid-refractory) if the serum levels of ALT, AST and IgG fail to normalize or worsen [3]. In AIH, the use of biologic therapy has been limited to this setting. Therapeutic recommendations beyond second-line therapy are yet to be made, but biologics do hold potential.

#### Rituximab and Infliximab

Rituximab is a chimeric monoclonal antibody against CD20. It depletes the B cells present in the peripheral blood, bone marrow and lymph nodes in experimental models, reduces the presentation of T cells, and improves AIH when refractory to the standard of care [139]. A retrospective multicenter experience with rituximab showed a biochemical improvement in all 22 treated patients with difficult-to-manage AIH [140]. The frequency of exacerbation after treatment was 23% during a median follow-up of 22 months, and no serious side effects were reported [140]. Infliximab is a recombinant monoclonal antibody that consists of the variable region of a mouse monoclonal antibody to TNF-α and the constant region of human IgG1. A study performed in patients with AIH (11 adults), who had failed to achieve remission with standard of care therapy and used infliximab as rescue treatment for at least six months, showed that biochemical remission was achieved by 60% of adults [141]. Seven patients developed infectious complications, two of them requiring hospitalizations; four had recurrent multiple infections, and three were withdrawn form therapy due to intolerance. Hepatoxicity is another complication in patients receiving infliximab (infliximab-induced DILI). It occurs in 1 of 120 patients, usually after four infusions, with about half of them requiring treatment with steroids; it resolves after the discontinuation of infliximab [142].

Therapy with rituximab and infliximab have been recognized as a possibility but further evidence-based medical research is required to warrant recommendation [139]. The use of biologic therapy for refractory AIH should be limited at this point to clinical trials that are well designed, appropriately powered, and controlled [139]. It should also be noted that a clinical trial with ianalumab, for the clarification of the roles of proliferation-inducing ligand (APRIL) and IL-2 in AIH, is on the way [139].

## 7. Conclusions

Autoimmune hepatitis (AIH) remains a diagnostic and therapeutic challenge. The implementation of immunosuppression with corticosteroids, with or without the use of azathioprine, has led to better outcomes and survival in patients with this condition. However, when intolerance or an inadequate response to treatment occurs, second-line therapy must be considered. The further implementation of randomized controlled trials is needed to assess and validate the use of other therapeutic options, such as biologic therapy.

## Figures and Tables

**Table 1 diagnostics-14-00382-t001:** Simplified criteria for autoimmune hepatitis.

Variable	Score
1. ANA or SMA/F-actin	
≥1:40	+1
≥1:80 or	+2
LKM ≥ 1:40 or	+2
SLA (+)	+2
2. Serum IgG	
> Upper limit of normality	+1
>1.1 × upper limit of normality	+2
3. Histologic findings	
Compatible with AIH	+1
Typical AIH	+2
4. Negative hepatitis viral markers	+2

Definite AIH is defined as ≥7 points, and probable AIH is defined as ≥6 points. Adapted from [101,102].

## Data Availability

Not applicable.

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
