# Peer review of "Autoimmune Hepatitis: A Diagnostic and Therapeutic Overview"

_diagnostics, 2024, doi:10.3390/diagnostics14040382_

Round 1

Reviewer 1 Report

Comments and Suggestions for Authors

Dear Colleagues,
The paper you wrote is of absolute interest and is relevant and informative. The relevance of the chosen topic is beyond doubt, since autoimmune hepatitis still remains one of the most unresolved hepatology problems .
The chosen approach to compiling a review on AIH seems rational and covers the main issues of both diagnosis and treatment of this diseases group.
There are no wishes or comments of a fundamental nature regarding the review text. All the main problem areas of differential diagnostics, diagnosis confirmation, first- and second-line therapy, as well as promising directions in the treatment of this autoimmune diseases group are touched upon.
The only thing I could wish for was a little bit more detail and more elucidation of modern approaches to the laboratory diagnosis of AIH, especially in the interpretation of laboratory examination results (lines 213-246). Maybe put the lists of main markers into a table and supplement the main text with it, since reading such a text overloaded with information is a little difficult?
This is purely a wish and has no fundamental character.
In conclusion, I can only congratulate you on a successful and interesting article on an important topic.

Author Response

Thank you for reviewing this manuscript.
In response to your suggestions, Table 1 was added within that section 5.4 (5.4 Diagnostic scoring systems) to allow a better understanding of how laboratory evaluation should be approached and integrated with histological and clinical findings.

Table 1. Simplified criteria for Autoimmune Hepatitis

Variable

Score

1. ANA or SMA/F-actin

≥1:40

+1

≥1:80 or

+2

LKM ≥1:40 or

+2

SLA (+)

+2

2. Serum IgG

> Upper limit of normality

+1

>1.1 x upper limit of normality

+2

3. Histologic findings

Compatible with AIH

+1

Typical AIH

+2

4. Negative hepatitis viral markers

+2

Table 1. Simplified criteria for Autoimmune Hepatitis. Definite AIH is defined as ≥7 points, probable AIH is defined as ≥6 points. Adapted from Galaski J, et al, J Hepatol 2021[101] and Flykshteyn, B, et al, Clin Liver Dis 2024 [102].

Reviewer 2 Report

Comments and Suggestions for Authors

-AIH diagnosis: discussing the diagnostic performances of the revised original AIH score (1999) and the simplified one (2008), the authors should also discuss the validation studies of the simplified AIH score in clinical practice in different geographical settings, as previously demonstrated (Validation of simplified diagnostic criteria for autoimmune hepatitis in Italian patients. Hepatology. 2009 May;49(5):1782-3; doi: 10.1002/hep.22825; Validation of the simplified criteria for diagnosis of autoimmune hepatitis in Chinese patients.  J Hepatol. 2011;54:340-7. doi: 10.1016/j.jhep.2010.06.032.).

-Epidemiology: the authors properly stated that "an older peak at onset has been reported in New Zealand and Denmark in >60 years old individuals". However, this observation has been previously reported also in Italy and other western countries. 

-An important topic is the impact of the different genetic background potentially explaining geographical differences in clinical and laboratory features of AIH, as previously demonstrated comparing Italian and North American AIH patients.

-The last point of clinical relevance and worth mentioning, is the risk of a severe complication of AIH. It is well recognized that AIH patients may develop “acute-on-chronic liver failure” (ACLF) either as hyperacute exacerbation of undiagnosed or misdiagnosed AIH, or in response to a second exogenous insult (viral, drug-induced, toxic) on typical AIH, possibly favored by long-term immunosuppression- It is therefore of clinical relevance to recall that, to mitigate the risk of ACLF in AIH patients, international practice guidelines call for vaccination against HAV and HBV at the time of diagnosis and the prudent use of any additional medication.

Author Response

Thank you for reviewing the manuscript.
In response to your comments, in blue text you will find the edits to the manuscript:

1. -"AIH diagnosis: discussing the diagnostic performances of the revised original AIH score (1999) and the simplified one (2008), the authors should also discuss the validation studies of the simplified AIH score in clinical practice in different geographical settings, as previously demonstrated (Validation of simplified diagnostic criteria for autoimmune hepatitis in Italian patients. Hepatology. 2009 May;49(5):1782-3; doi: 10.1002/hep.22825; Validation of the simplified criteria for diagnosis of autoimmune hepatitis in Chinese patients.  J Hepatol. 2011;54:340-7. doi: 10.1016/j.jhep.2010.06.032.)."
This section was completely modified to include relevant studies regarding the criteria validation in multiple regions and the simplified criteria for AIH was added.

5.4. Diagnostic scoring systems.

The International Autoimmune Hepatitis Group (IAIHG) created a diagnostic scoring system [94], later revised to aid in the diagnosis of AIH [67], and a simplified version was subsequently created [95]. The revised scoring system exhibits a greater sensitivity for AIH when compared to the simplified scoring system (100% versus 95%), while the simplified version has greater specificity (90% versus 73%) and accuracy (92% versus 82%) [96]. These score’s diagnostic performance has been tested in different geographic settings. Due to its clinical applicability, the simplified system was tested in cohorts including population from Germany, Japan, Spain, Greece, Norway, Brazil, Austria, United Kingdom, and USA, and validated to populations including viral, metabolic, toxic, cholestatic and genetic hepatic syndromes, with a sensitivity of 99% and specificity of 97% [70, 95]. A validation study from a chinese cohort reported that for the simplified score the sensitivity and specificity was 90% and 95%, respectively [97]. Likewise, an Italian study reported an overall sensitivity and specificity of 91% and 94%, respectively [98].

The revised criteria scoring system has been used in clinical practice to assess patients with few or atypical features of AIH [99]. However, although it is very extensive and complex, it fails to distinguish AIH from cholestatic syndromes.

Conversely, the simplified scoring system (table 1) is based on four parameters: presence and titer of autoantibodies detected by IFL and ELISA (for anti-SLA/LP), concentration of serum IgG, presence of typical or compatible histology, and absence of viral hepatitis markers [70]. Although the simplified criteria work for AIH-PBC and are useful in excluding patients with AIH from patients with other conditions and concurrent immune features, they are more likely to exclude atypical cases of AIH [95] and fail to grade responses to corticosteroid treatment [100].

Given the aforementioned considerations, clinical judgment is crucial for the use of these scoring systems.

Table 1. Simplified criteria for Autoimmune Hepatitis

Variable

Score

1. ANA or SMA/F-actin

≥1:40

+1

≥1:80 or

+2

LKM ≥1:40 or

+2

SLA (+)

+2

2. Serum IgG

> Upper limit of normality

+1

>1.1 x upper limit of normality

+2

3. Histologic findings

Compatible with AIH

+1

Typical AIH

+2

4. Negative hepatitis viral markers

+2

Table 1. Simplified criteria for Autoimmune Hepatitis. Definite AIH is defined as ≥7 points, probable AIH is defined as ≥6 points. Adapted from Galaski J, et al, J Hepatol 2021[101] and Flykshteyn, B, et al, Clin Liver Dis 2024 [102].

2.- "Epidemiology: the authors properly stated that "an older peak at onset has been reported in New Zealand and Denmark in >60 years old individuals". However, this observation has been previously reported also in Italy and other western countries." This section was also modified and now includes relevant new epidemiological studies that better reflect current global AIH epidemiology understanding.

AIH occurs within all ages and ethnic groups, and interestingly, clinical expression seems to vary depending on race and ethnicity. AIH is more predominant among adult females (71%-95%) [5] and girls (60%-76%)[6]. Age peaks have been typically described at 10-30 and 40-60 years [7], nonetheless, in recent years an older peak at onset has been reported almost universally, mainly considering western Europe and North American populations, in >60 years old individuals. [8-10]. At disease onset, high frequency of icteric disease seems to be present in Alaskan natives, and more severe disease in North American Aboriginal/First Nations [11]. In Hispanic population with AIH, belligerent biochemical and histological features are often seen as well as a high prevalence of cholestatic disease [12]. Cirrhosis is commonly found among Hispanic [13] and African American populations with accelerated progression of the disease; the latter is a group that has also shown a high rate of recurrence after LT, along with higher mortality than white American patients [14]. The incidence of AIH varies world-wide and mainly depends on region and age at onset. A marked increase in incidence of nearly 50% has been reported over the past decades in Denmark, Sweden, Spain, and the Netherlands [5, 7, 15]. Moreover, recent large epidemiological studies including populations from five continents (although including mostly western world countries), have reported changes in incidence and prevalence, with AIH affecting mainly countries with high human development index, adults aged over 65 years old, high latitude (>45 degrees), and North American populations [9, 10].

3. "-An important topic is the impact of the different genetic background potentially explaining geographical differences in clinical and laboratory features of AIH, as previously demonstrated comparing Italian and North American AIH patients." This section (2.1) has been modified so that now it includes relevant geographic differences regarding genetics, and its clinical expression.

2.1. Genetic susceptibility

HLA A1-B8-DR3 and HLA DR4 haplotypes have long been strongly linked to AIH, with 84% of AIH patients from western Europe and North America having these haplotypes. Moreover, patients from Japan (DRB1*0405), Mexico (DRB1*0404), Argentina (HLA-DRB1*0301, DRB1*0405) and Brazil (DRB1*13 and DRB1*03) have a particular HLA-risk profile [16, 17]. Also, the A1-B8-DR3 haplotype has been linked to more severe disease, need for liver transplantation, less response to steroids, and more relapse after transplant [18].

The differences seen across populations might reflect genetic predispositions, along with different immunologic and environmental influences [3], however, it’s highly likely that socioeconomic factors (delivery of healthcare, delayed diagnosis, competing risk factors) also play an important role in the pathogenesis.    

4. "-The last point of clinical relevance and worth mentioning, is the risk of a severe complication of AIH. It is well recognized that AIH patients may develop “acute-on-chronic liver failure” (ACLF) either as hyperacute exacerbation of undiagnosed or misdiagnosed AIH, or in response to a second exogenous insult (viral, drug-induced, toxic) on typical AIH, possibly favored by long-term immunosuppression- It is therefore of clinical relevance to recall that, to mitigate the risk of ACLF in AIH patients, international practice guidelines call for vaccination against HAV and HBV at the time of diagnosis and the prudent use of any additional medication." The specifics of acute AIH and ALF associated-acute liver failure were included in section 4.7:

4.7. Acute disease, acute liver failure and chronic complications in AIH.

AIH is one of the most common causes of acute liver failure (ALF) in the USA. According to AASLD, acute severe AIH is defined as an acute liver injury (ALI) with jaundice plus INR >1.5 and <2.0 [3]. As mentioned earlier, acute AIH is a frequent form of presentation [24]. Around 6% of patients develop ALF according to data from European and North American patients [58]. Recent data from a US cohort reported that 16% of patients with AIH-associated ALI subsequently developed ALF. At presentation, ALT was 449 (median, 227–805), bilirubin was 22.8 (median , 17.9–28.0). Only 70% had antinuclear antibodies (ANA) ≥ 1:40 and 43% had anti-smooth muscle antibodies (ASMA) ≥ 1:40. At 21 days, transplant-free survival was 15% and 24% died without liver transplant.

In the acute AIH group, ANA and ASMA have been reported as negative in approximately 11%, and immunoglobulin levels were also found normal in 15-39% [59, 60]. Moreover, the simplified AIH diagnostic score has been shown to be less sensitive than the original revised version [61].

Liver biopsy should be performed if possible. Histopathologic findings in this context might include lobular hepatitis, lymphoplasmacytic infiltrate, and interface hepatitis support acute AIH. Specific ALF features encompass central perivenulitis, lymphoplasmacytic infiltrate, lymphoid follicles, and massive hepatic necrosis [58]. Heterogeneous hypoattenuated regions within the liver can be seen in an unenhanced computer tomography in 65% of patients with acute severe AIH [62]. Thus, this population remains a diagnostic challenge due to discrepancies in laboratory and histological phenotypes compared to classical AIH.

Reviewer 3 Report

Comments and Suggestions for Authors

In this review, the authors presented and discussed important issues of autoimmune hepatitis diagnosis and management.

The manuscript is well organized and discussed. However, I have minor comments that should be addressed. 

1) Autoantibodies: the authors stated that " Antibodies against filamentous (F) actin (antiactin) are a subset of ASMA and are present in 86%-100% of patients with AIH and ASMA [68, 69]". In my opinion, this is a very important point because it has been also demonstrated that anti-F-actin antibodies are more reliably detected by indirect immunofluorescence that, although with the limit of the operator-dependent assessment, has lower false-positive cases compared with the ELISA assay, as previously demonstrated (Antibodies to filamentous actin (F-actin) in type 1 autoimmune hepatitis. J Clin Pathol. 2006 Mar;59(3):280-4. doi: 10.1136/jcp.2005.027367). This would further support the proper statement of the authors that "Increase in training and expertise is essential for laboratory personnel and clinicians to perform a correct interpretation of results." 

2) the AIH with "acute presentation" is a clinically very important topic. In such an AIH patients subgroup, it has been recently demonstrated that, although previously validated in real-world studies, the simplified AIH diagnostic scoring system performs less than the original revised one in AIH patients with acute presentation, because of the higher rate of patients having a normal IgG serum levels and a lower frequency of autoantibody positivity as recently demonstrated (Limitation of the simplified scoring system for the diagnosis of autoimmune Hepatitis with acute onset. Liver Int. 2021 Mar;41(3):529-534. doi: 10.1111/liv.14778.).

Author Response

Thank you for reviewing the manuscript:

In response to your comments, below you will find our edits:

1) Autoantibodies: the authors stated that " Antibodies against filamentous (F) actin (antiactin) are a subset of ASMA and are present in 86%-100% of patients with AIH and ASMA [68, 69]". In my opinion, this is a very important point because it has been also demonstrated that anti-F-actin antibodies are more reliably detected by indirect immunofluorescence that, although with the limit of the operator-dependent assessment, has lower false-positive cases compared with the ELISA assay, as previously demonstrated (Antibodies to filamentous actin (F-actin) in type 1 autoimmune hepatitis. J Clin Pathol. 2006 Mar;59(3):280-4. doi: 10.1136/jcp.2005.027367). This would further support the proper statement of the authors that "Increase in training and expertise is essential for laboratory personnel and clinicians to perform a correct interpretation of results."
The cited paragraph was edited as follows:
Antibodies against filamentous (F) actin (antiactin) are a subset of ASMA and are present in 86%-100% of patients with AIH and ASMA [78, 79] Interestingly, it has been shown that the detection of anti-F-actin antibodies is more consistently achieved through indirect immunofluorescence (observer’s experience-dependent) compared to the ELISA assay [80].

2) the AIH with "acute presentation" is a clinically very important topic. In such an AIH patients subgroup, it has been recently demonstrated that, although previously validated in real-world studies, the simplified AIH diagnostic scoring system performs less than the original revised one in AIH patients with acute presentation, because of the higher rate of patients having a normal IgG serum levels and a lower frequency of autoantibody positivity as recently demonstrated (Limitation of the simplified scoring system for the diagnosis of autoimmune Hepatitis with acute onset. Liver Int. 2021 Mar;41(3):529-534. doi: 10.1111/liv.14778.). The specifics of acute AIH and AIH associated-acute liver failure were included in section 4.7: 

4.7. Acute disease, acute liver failure and chronic complications in AIH.

AIH is one of the most common causes of acute liver failure (ALF) in the USA. According to AASLD, acute severe AIH is defined as an acute liver injury (ALI) with jaundice plus INR >1.5 and <2.0 [3]. As mentioned earlier, acute AIH is a frequent form of presentation [24]. Around 6% of patients develop ALF according to data from European and North American patients [58]. Recent data from a US cohort reported that 16% of patients with AIH-associated ALI subsequently developed ALF. At presentation, ALT was 449 (median, 227–805), bilirubin was 22.8 (median, 17.9–28.0). Only 70% had antinuclear antibodies (ANA) ≥ 1:40 and 43% had anti-smooth muscle antibodies (ASMA) ≥ 1:40. At 21 days, transplant-free survival was 15% and 24% died without liver transplant.

In the acute AIH group, ANA and ASMA have been reported as negative in approximately 11%, and immunoglobulin levels were also found normal in 15-39% [59, 60]. Moreover, the simplified AIH diagnostic score has been shown to be less sensitive than the original revised version [61].

Liver biopsy should be performed if possible. Histopathologic findings in this context might include lobular hepatitis, lymphoplasmacytic infiltrate, and interface hepatitis support acute AIH. Specific ALF features encompass central perivenulitis, lymphoplasmacytic infiltrate, lymphoid follicles, and massive hepatic necrosis [58]. Heterogeneous hypoattenuated regions within the liver can be seen in an unenhanced computer tomography in 65% of patients with acute severe AIH [62]. Thus, this population remains a diagnostic challenge due to discrepancies in laboratory and histological phenotypes compared to classical AIH.

Round 2

Reviewer 2 Report

Comments and Suggestions for Authors

The revised version satisfactorily addresses the raised points and the manuscript can be now accepted as it is.